# Impact of a Fish-Based Restrictive Ketogenic Diet on Body Composition and Strength Capacity: A Pre–Post Study

**DOI:** 10.3390/nu17081297

**Published:** 2025-04-08

**Authors:** Katarzyna Siedzik, Kamil Góral, Ewa Rodziewicz-Flis, Robert A. Olek, Wiesław Ziółkowski

**Affiliations:** 1Doctoral School, Poznan University of Physical Education, 61-871 Poznan, Poland; 2Department of Physiotherapy, Gdansk University of Physical Education and Sport, 80-336 Gdansk, Poland; 3Department of Athletics, Strength, and Conditioning, Poznan University of Physical Education, 61-871 Poznan, Poland; 4Department of Rehabilitation Medicine, Faculty of Health Sciences, Medical University of Gdansk, 80-219 Gdansk, Poland

**Keywords:** body composition, isometric muscle strength, ketogenic diet, nutritional intervention, omega-3, somatic disorders, weight loss

## Abstract

**Background**: The ketogenic diet (KD) is becoming a popular nutritional model for athletes. One limitation of this diet is the use of animal-meat products, which may be unacceptable to some people. Moreover, the relationship between this diet’s use, body composition, and strength capacity has not been thoroughly investigated. This study aimed to assess the effects of a two-week fish-based restrictive KD on body composition, strength capacity (isometric muscle strength), and somatic disorders in healthy adults. **Methods:** Participants were recruited through advertisements, posters displayed at the university, and information shared among colleagues. Ultimately, 14 individuals qualified for this study. For two weeks, they followed a designated ketogenic diet enriched with fish and omega-3 fatty acids, with a calorie reduction of −500 kcal/day. The study was designed and conducted as a single-group pre–post study. Before and after completing the diet, participants’ body composition (body mass, body fat, fat-free mass, and visceral fat) and strength capacity (knee extensor peak torque [KEPT] and knee flexor peak torque [KFPT]) were measured. The survey also analyzed participants’ somatic disorders such as nausea, vomiting, energy level, diarrhea, constipation, drowsiness, sleep problems, hunger, thirst, and bad breath. The average results of both trials obtained before and after the diet were compared using the paired Student’s *t*-test or non-parametric test. The significance level was set at 0.05. **Results:** After two weeks of a fish-based restrictive KD, significant increases in ketone bodies were observed in both the blood (pre: 0.21 mmol/L ± 0.15 vs. post: 2.20 ± 1.19, *p* < 0.001) and urine (pre: 0.00 mmol/L ± 0.00 vs. post: 4.64 mmol/L ± 3.24, *p* < 0.001). Body composition changes included significant reductions in total body mass (pre: 76.66 kg ± 11.06 vs. post: 73.77 kg ± 10.76, *p* < 0.001), fat mass (pre: 21.34 kg ± 7.36 vs. post: 19.73 kg ± 7.20, *p* < 0.001), and visceral fat (pre: 708.50 g ± 627.67 vs. post: 603.21 g ± 567.82, *p* = 0.0012). Skeletal muscle mass decreased by 2% (pre: 27.75 kg ± 5.80 vs. post: 27.03 kg ± 5.47, *p* = 0.01), though no significant changes were seen in muscle strength when adjusting for body mass or skeletal muscle mass. No major adverse effects were noted in relation to somatic disorders, although some participants reported increased thirst and bad breath. **Conclusions**: After 14 days of a fish-based restrictive KD, a significant reduction in body mass and changes in body composition were observed, with no loss of muscle strength. This type of diet may serve as an effective method for rapid body mass reduction in sports with weight categories, while preserving muscle strength, which is crucial for athletes. It could also be an alternative means for individuals to eliminate animal meat from their diet.

## 1. Introduction

The ketogenic diet (KD) and the low-carbohydrate diet (LCD) have been among the most popular dietary interventions in recent years. In both interventions, it is essential to limit carbohydrates (CHO) in favor of fats, and consequently their metabolites and ketone bodies [1]. During ketogenesis, three types of ketone bodies are produced: acetoacetate (AcAc), acetone, and the predominantly produced 3-hydroxybutyrate (BHB) [2]. Energy production is related to the oxidation of carbohydrates (CHO) and fat. CHO is stored in the muscles and liver as glycogen (untrained individuals present level of only ~8000 kJ (approximately 1900 kcal), while these levels are approximately 20–50% higher in trained men and women) [3]. Unlike carbohydrates, fat is a less limited energy source. Therefore, using carbohydrate stores during KD causes the metabolism to switch to the oxidation of fatty acids and ketone bodies, which promotes body mass loss [3,4,5]. This information is of interest for this type of diet [6].

Moreover, it should be noted that the desired effects can be achieved with a restrictive diet [7]. The expected reduction in dietary energy intake should be approximately 500 kcal/day lower than the daily energy requirement [7]. However, caloric restriction causes, in addition to benefits such as body mass loss, adverse effects such as the loss of muscle mass and an associated decrease in skeletal muscle strength [8,9,10], slower metabolism [11], and feelings of increased hunger [12]. Increased protein supply in the ketogenic diet is thought to ameliorate adverse effects and consequently intensify the production of ketone bodies that reduce hunger [12]. However, for some athletes and people who want to lose body mass, a KD based on products containing large amounts of meat, due to their own conscious decision, is unacceptable. This study presents an alternative ketogenic diet that replaces meat products (like poultry, beef, pork, duck, lamb, etc.) with fish and other products rich in omega-3 fatty acids. Fish is a very good source of protein, fat (especially omega-3 acids), vitamins D, A, and E, and micro- and macronutrients including iodine, selenium, iron, potassium, magnesium, and zinc [13,14].

The most valuable fatty acids include polyunsaturated fatty acids (PUFA), particularly omega-3 fatty acids. In products of plant origin, they are mainly found as α-linolenic acid (ALA), and in fish, they are found as eicosapentaenoic- acid (EPA) and docosahexaenoic acid (DHA). EPA is a polyunsaturated fatty acid with a 20-carbon chain and five cis double bonds. EPA is known for its anti-inflammatory properties, acting as a precursor for prostaglandin-3, thromboxane-3, and leukotriene-5 eicosanoids, which play roles in reducing inflammation and preventing blood clotting. It is primarily found in cold-water fish like salmon, mackerel, and sardines, and can also be obtained from fish oil supplements. EPA is used to lower triglyceride levels in the blood, reduce the risk of heart disease, and alleviate symptoms of depression [15].

DHA is another polyunsaturated fatty acid, but with a 22-carbon chain and six cis double bonds. DHA is crucial for brain health, supporting cognitive function and eye health. It is a major component of neuronal membranes and helps in the optimal functioning of neuronal membrane proteins. DHA is found in fatty fish like salmon, herring, and mackerel, as well as in fish oil and algae oil. It is important for brain development, particularly in infants, and also supports heart health, potentially reducing the risk of chronic diseases [16]. EPA and DHA have primary anti-inflammatory effects [17]. Additionally, research suggests that omega-3 PUFAs may help to reduce or alleviate the symptoms of depression, polycystic ovary syndrome (PCOS), autoimmune diseases (including Hashimoto’s disease, rheumatoid arthritis, psoriasis, and vitiligo), acne, and insulin resistance [4,18,19]. Omega-3 acids are associated with reduced inflammation and are beneficial in autoimmune diseases and the immune system response [19]. Moreover, omega-3 has been clinically linked to reducing inflammation in oncological treatments and has shown promise in recent studies on the effect of omega-3 in patients undergoing bone marrow transplantation [20,21].

In contrast, it is postulated that omega-3 fatty acids facilitate insulin-sensitive protein anabolism through the Akt-mTOR-S6K1 pathway, which prevents anabolic resistance and leads to decreased muscle atrophy induced by disuse [5]. Furthermore, another study reported that ingesting 5% fish oil reduced disuse muscle atrophy via the Akt pathway through E3 ubiquitin ligases and the S6K1 pathway [22]. This allowed us to cautiously suggest that a KD which is rich in fish and omega-3 fatty acids could be beneficial in inhibiting the loss of muscle mass following a restrictive diet.

Therefore, this alternative KD seems to be a favorable choice for athletes in the context of their health (not as a ‘standard’ ketogenic diet rich in saturated fatty acids) and exercise capacity [23,24].

Moreover, an adequate supply of protein in the ketogenic diet should be considered so that excess protein does not lead to the production of physiological glucose from amino acids during gluconeogenesis [25]. According to the recommendations, the range for the protein supply should be between 1.4 and 1.6 g protein/kg of body mass, which is usually approximately 15% of the daily energy supply [24,25,26,27,28,29].

While previous studies highlight the beneficial effects of omega-3 fatty acids, particularly EPA and DHA, on inflammation, insulin resistance, and muscle anabolism, there remains a lack of research focusing on the effects of a KD rich in these omega-3 fatty acids, specifically from fish, on body composition and muscle mass in a short-term context. Most of the existing literature on the KD emphasizes its impact on weight loss and fat reduction, but there is limited evidence regarding its influence on muscle preservation, especially when combined with a restrictive diet.

We hypothesized that using this alternative KD would reduce body mass and fat mass and at the same time would not cause a decrease in skeletal muscle strength or significant changes in somatic disorders.

This study aims to fill this gap by examining the effects of a fish-based restrictive KD on body mass, body composition, and skeletal muscle strength over a 14-day period, providing insights into its potential benefits for athletes and individuals seeking rapid body mass reduction without significant muscle loss. This study aimed to assess the effects of a two-week fish-based restrictive KD on body composition, strength capacity (isometric muscle strength), and somatic disorders in healthy adults.

## 2. Materials and Methods

### 2.1. Sample Size

The previously reported results [7] were used for the dependent sample’s *t*-test sample size calculation, performed using Statistica 13.1 software (Dell Inc., Tulsa, OK, USA).

Means and standard deviations (SDs) of body mass were applied with the alpha level and power of the test set at 0.05 and 0.95, respectively. Based on these parameters, the required sample size obtained was equal to 12.

### 2.2. Participants

The study was conducted with recruitment carried out through advertisements, posters displayed at the university, and information shared among colleagues. Efforts were made to ensure diversity among participants, resulting in a group with varying body weights, genders, and levels of physical activity. Fourteen healthy participants (age 45 ± 8 years; height 174 ± 10 cm; body mass 76.7 ± 11.1 kg, female *n* = 5, male *n* = 9) were recruited for this study by self-selection (20 people were recruited, and finally 14 people were enrolled in the study). Data were collected for all participants (*n* = 14), except for isometric muscle strength (*n* = 13). The participants included both active and sedentary individuals, and the effects of the ketogenic diet were observed consistently across both groups, enhancing the applicability of the results to a wider population.

### 2.3. Eligibility Criteria

The inclusion criteria were those who were female or male, non-smokers, drank alcohol occasionally, and had not having performed physical exercise for at least 48 h before the examination. The exclusion criteria included cardiovascular, thyroid, gastrointestinal, respiratory, and other metabolic diseases; adherence to special diets; nutritional supplements; and medication to control blood lipids or glucose. During the study protocol, the participants were asked to stop their physical exercise. While this study was conducted, no specific cultural or geographical dietary influences were identified that could impact the findings.

### 2.4. Ethics

All subjects read and signed an informed consent document with the description of the testing procedures approved by the Independent Bioethics Committee for Scientific Research, District Medical Chamber in Gdansk (KB 27-19), that conformed to the standards for the use of human subjects in research, as outlined in the Declaration of Helsinki, Clinical Trial registration number NCT05558488.

### 2.5. Study Design and Procedures

This study was designed as a one-sample pre–post study, where the same group of participants was assessed before and after a 14-day dietary intervention. This design allows for evaluating within-subject changes over time, minimizing inter-individual variability. By comparing baseline and post-intervention measurements, this study aimed to determine the effects of a fish-based ketogenic diet. The pre–post approach is particularly useful for short-term dietary interventions, as it provides insights into rapid physiological adaptations without the need for a control group.

The participants were invited to a meeting with the experiment director before the start of the research, where they were trained in detail regarding the assumptions of the experiment, its course, the principles of the ketogenic diet, and the observation of symptoms.

Measurements were performed the day before the tests were started and after 14 days. A 14-day time frame is sufficient to observe the initial physiological effects of a ketogenic diet, including reductions in total body mass and visceral fat, which typically occur within the first two weeks. This study focused on short-term changes, making this duration ideal for assessing rapid body composition shifts. Additionally, the minimal reduction in skeletal muscle mass and the preserved isometric muscle strength further indicate that 14 days is an appropriate time period to evaluate these early outcomes, especially for athletes seeking quick results. Measurements were performed by the same operator under the same conditions. The study design is illustrated in Figure 1.

### 2.6. Dietary Intervention

Each participant received a nutritional plan that was the same qualitatively but differed quantitatively; a 500 kcal reduction based on the Estimated Energy Requirement (EER) was assumed. It was calculated for each individual based on the Mifflin–St Jeor equation (basal metabolic rate) formula [30]:Men: (10 × weight in kg) + (6.25 × height in cm) − (5 × age in years) + 5Women: (10 × weight in kg) + (6.25 × height in cm) − (5 × age in years) − 161

Subsequently, physical activity level (PAL) was used to assess individuals’ physical activity patterns and energy expenditure for placement into one of four PAL categories: sedentary, low active, active, or very active. This helped us to calculate the EER and made an individual reduction (minus 500 kcal/day). Before the start of this study, each participant obtained information about the basic principles of the ketogenic diet and about the allowed and prohibited products. The food lists encouraged the consumption of fish (fish included in the plan were cod, mackerel, pollock, tuna, and salmon), raw and cooked vegetables, eggs, fruits with the lowest glycemic index (blueberries and raspberries), plant oils and fats from avocadoes, and olives. The drinks permitted were tea and black coffee without sugar, and the foods and beverages to be avoided were alcohol, meat (any meat), bread, pasta, rice, milk, dairy, and potatoes. The diet was designed to be isoproteic (1.6 g·kg^−1^·body mass^−1^·day^−1^), with three meals daily. The distribution of macronutrients in the ketogenic diet was as follows: protein, 1.6 g·kg^−1^·body mass^−1^·day^−1^ (~20%), fats (~75%, with a strong emphasis on the content of omega-3 fatty acids), and carbohydrates (<30 g·day^−1^; <10%) (Table A1). The subjects consumed 1500 to 2700 kcal, depending on the daily energy requirement. Each study participant received their own individual diet plan—where the energy value and the share of individual macronutrients (proteins, fats, carbohydrates) were individually counted—as presented above. The participants were informed that during the experiment they must completely follow the diet they received individually, while carefully monitoring the weight of the products they consumed—that is, they were required to eat all meals, not snack on anything additional, and keep an eye on the prescribed portions of each product. The energy value and content of dietary macronutrients, micronutrients, vitamins, and unsaturated fatty acids, omega-3 and omega-6, depending on the daily energy requirement of the subjects, are shown in Table A1. The person who carried out the dietary analysis was someone who had completed both a bachelor’s and a master’s degree in the field of dietetics.

### 2.7. Body Composition

Body composition analysis was performed using precise dual-energy X-ray absorptiometry (GE Healthcare Prodigy Primo) at baseline (the day before the start of the ketogenic diet) and the day after completion of the ketogenic diet. Body mass, fat-free mass (FFM), body fat mass (FM), and visceral fat (VFA) were assessed. Body composition analysis was performed in a fasting state 12 h after the last meal and during drinking between 7:00 and 9:00 a.m. in the morning after blood collection. For the measurement, the participants were asked to remove jewelry and put away their electronic devices. The measurements were performed in a lying position and the participants were dressed only in their underwear. A qualified physician performed all the scans. All measurements were performed by the same operator before and after the study according to standard procedures.

### 2.8. Skeletal Muscle Mass Analysis

Muscle mass (MM) was calculated based on appendicular lean soft tissue (ALST), which is the sum of the mass of soft tissues of the upper and lower limbs obtained by the DXA method. To determine muscle mass, data were collected using precise dual-energy X-ray absorptiometry (GE Healthcare Prodigy Primo).

It was calculated for each individual using the following equation [31]:muscle mass (MM) = (1.13 × ALST) − (0.02 × age in years) + (0.61 × sex) + 0.97

### 2.9. Isometric Muscle Strength Test

The maximal isometric knee extensor torque at 90° was measured using a Biodex Dynamometer (Biodex Medical Systems, Inc., Shirley, NY, USA). Before the study participants were seated on the dynamometer, they were asked to warm up for 5 min at 100 W on a cycloergometer (Monark, Vansbro, Sweden). All seat settings were set individually and repeated during the next trial after the dietary intervention. To avoid movement of the trunk and pelvis during the test, stabilizing belts were carefully fastened, and the participants were asked to place their arms crossed on their chest. According to the manufacturer’s recommendations, the lateral epicondyle of the knee joint was the axis of the measured ranges of motions during the main measurements of the study. The time of a single set of maximum knee extensors and flexor contraction was four seconds, with a twenty second pause between each trial. Three sets of data were generated in each experiment. Verbal encouragement was provided during the trials. The highest and mean values were used for statistical analysis.

The strength was measured in Nm. As explained for the force measurement, the torque and knee-angle data were transferred to a PC. Data showing the highest torque for each measurement (PC with Windows 10 Enterprise LTSC, Advantage BX ™ Software 5.1.0 (Biodex Medical Systems, Inc., Shirley, NY, USA)) were used for further analysis.

### 2.10. Ketone Bodies in Blood and Urine

The concentration of ketone bodies (β-hydroxbutyrate (βHB)) in the capillary blood was measured using a Freestyle Libre OptiumXido Neo glucometer (Abbott, Abbott Diabetes Care Ltd., Witney, Oxon, UK) and OptiumXido β-ketone testing strips (Abbott Diabetes Care Ltd., Witney, Oxon, UK). The concentration of ketone bodies (acetoacetic acid) in the urine was measured using a strip test (Keto-Diastix, Bayer Consumer Care AG, Basel, Switzerland). Fingertip blood and urine samples were collected from the patients pre and post KD. All blood samples were obtained from the patients in a seated upright position by a qualified physician.

### 2.11. Somatic Test

A Likert scale was used to assess somatic disorders. It is a psychometric scale that is commonly used in research that employs questionnaires. The format of the typical five-level Likert element used in these studies is as follows: 1, I categorically disagree; 2, I disagree; 3, I neither agree nor disagree; 4, I agree; and 5, I definitely agree (Table A2).

The Likert scale used to measure somatic symptoms in this study was specifically developed for this research. It is a custom-designed psychometric tool created to assess various somatic disorders including nausea, vomiting, energy level, diarrhea, constipation, concentration disorders, drowsiness, sleep problems, hunger, thirst, and bad breath. As it was newly developed for this study, it has not undergone formal validation or reliability testing.

Since the scale was constructed to capture a wide range of somatic symptoms associated with the ketogenic diet, its psychometric properties (reliability and validity) were not formally assessed in this study. However, it was designed to follow standard Likert scale procedures and was reviewed to ensure that the questions were relevant and comprehensive. The survey questions concerned nausea, vomiting, energy level, diarrhea, constipation, concentration disorders, drowsiness, sleep problems, hunger, thirst, and bad breath.

All participants responded to this survey before and 14 days after starting the nutritional protocol according to their feelings on the above scale.

### 2.12. Statistical Analysis

Results are presented as the mean ± standard deviation (SD). Normality was analyzed using the Shapiro–Wilk test, and statistical analyses were performed using the paired Student’s *t*-test or non-parametric test (Statistica 13.1 software (Dell Inc., Tulsa, OK, USA)). The significance level was set at a standard value of 0.05.

Graphical results are presented using GraphPad Prism 10 (GraphPad Software, Boston, MA, USA, 2024).

## 3. Results

Due to the limited supply of sugars in the diet, KD-induced physiological ketosis, characterized by the increased production of ketone bodies in the blood and urine, was observed in the subjects (Figure 2).

After two weeks of a fish-based restrictive KD, a significant increase in ketone bodies in the blood (pre: 0.21 mmol/L ± 0.15 vs. post: 2.20 ± 1.19, *p* < 0.001, Figure 2A), in the urine (pre: 0.00 mmol/L ± 0.00 vs. post: 4.64 mmol/L ± 3.24, *p* < 0.001, Figure 2B) was recorded.

Changes in body composition are summarized in Figure 3.

As shown in Figure 3, after two weeks of a fish-based restrictive KD, a significant decrease in the body mass (pre: 76.66 kg ± 11.06 vs. post: 73.77 kg ± 10.76, *p* < 0.001, Figure 3A), in the body fat (pre: 21.34 kg ± 7.36 vs. post: 19.73 kg ± 7.20, *p* < 0.001, Figure 3B), in the percentage of body fat (pre: 28.69% ± 8.45 vs. post: 27.60% ± 8.50, *p* = 0.0019, Figure 3C), in the visceral fat (pre: 708.50 g ± 627.67 vs. post: 603.21 g ± 567.82, *p* = 0.0012, Figure 3D), and in the fat-free mass (pre: 52.51 kg ± 9.41 vs. post: 50.99 kg ± 8.80, *p* = 0.0017, Figure 3E) was recorded. There was no significant change in FFM (%) (pre: 68.60% ± 7.95 vs. post: 69.25% ± 7.52, *p* = 0.064, Figure 3F).

Maintaining skeletal muscle mass is critical in KD, especially in athletes, to determine the success of their diet. The effect of diet on skeletal muscle loss was determined (Figure 4A).

After two weeks of a fish-based restrictive KD, a significant decrease (*p* = 0.01) in muscle mass (pre: 27.75 kg ± 5.80 vs. post: 27.03 kg ± 5.47) was noted (Figure 4A).

Considering the changes in body mass composition and skeletal muscle mass, it would be interesting to investigate the effects of KD on skeletal muscle strength. The results of these measurements are summarized in Figure 4B and Table 1 and Table 2.

As shown in Figure 4B, the mean value of the average isometric strength of the extensor thigh muscles was significantly lower after two weeks of KD, 221.16 ± 57.08 Nm vs. 228.42 ± 58.04 Nm after and before the diet, respectively, (*p* = 0.0256). There were no significant differences in the knee extensor peak torque (pre: 236.53 ± 61.12 Nm vs. post: 221.10 ± 46.60 Nm, *p* = 0.0869), as well as in the average (pre: 91.25 ± 25.04 Nm vs. post: 88.76 ± 28.25 Nm, *p* = 0.2793) and peak (pre: 97.75 ± 29.93 Nm vs. post: 94.55 ± 30.45 Nm, *p* = 0.1563) knee flexor torque.

However, no significant differences were observed when isometric strength values were expressed per kg of body mass (Table 1) or skeletal muscle mass (Table 2).

There were no significant differences between before and after two weeks of a fish-based restrictive KD (*p* > 0.05) in the flexor peak, flexor average, extensor peak, and extensor average strength of knee torque.

As shown in Table 2, no significant differences were noted between before and after two weeks of a fish-based restrictive KD (*p* > 0.05) in any of the parameters.

Another essential element of the diet is the assessment of somatic sensations and disorders that accompany its use in humans. However, there were no significant differences in diet-related sensations, such as nausea, vomiting, energy levels, diarrhea, constipation, drowsiness, sleep problems, hunger, thirst, or bad breath. Table 3 presents the results of the survey.

Somatic disorders were assessed in subjects (*n* = 14) before (pre) and after (post) the two-week KD. Values (pre and post) are expressed as the mean ± standard deviation. No significant differences were found between the mean pre and post results (*p* < 0.05). Improvement refers to the number of people whose results improved after KD and deterioration refers to the number of people whose results deteriorated after KD.

There were no significant differences in the mean values between before and after two weeks of a fish-based restrictive KD (*p* > 0.05) in somatic disorders. However, interestingly, apart from bad breath and increased thirst ratings, in which the deterioration of results was observed, beneficial effects were noted in a more significant number of subjects.

## 4. Discussion

Here, we report that a two-week fish-based restrictive KD influences body mass and composition without significantly affecting somatic disorders or strength capacity in healthy adults. While the absence of significant adverse somatic effects in our study is a positive finding, it is important to note that the tolerability of this no-meat-except-fish restrictive ketogenic diet could differ when compared to standard ketogenic diets. Standard ketogenic diets, which typically include higher amounts of saturated fats, may elicit a different range of somatic symptoms, such as gastrointestinal discomfort, bad breath, and changes in energy levels. Our study showed no significant differences in most somatic sensations after the two-week diet period, except for an increase in thirst and bad breath, which could be specific to the modified ketogenic approach used here.

### 4.1. KD, Rich in Fish and Omega-3 Acids, Induces Ketosis

It was assumed that this alternative ketogenic diet rich in fish and omega-3 fatty acids, owing to the limited amount of carbohydrates, would put subjects in a state of physiological ketosis, as evidenced by the increased concentration of ketone bodies in the blood and urine. Similar effects were observed in studies using a high-fat ketogenic diet, in which the energy intake corresponded to 13% and 74% of total energy for carbohydrates and fat, respectively. This diet is based on animal fats and animal fat-rich products, including pork, beef, butter, lard, cheese, eggs, and vegetables at up to 100 g/day [7]. KD induces ketosis that is not pathological but is a physiological condition [28]. Under normal conditions, the concentration of ketone bodies is lower (<0.3 mmol/L) compared to glucose (~=4 mmol) [32,33]. The Michaelis constants (Km) of enzymes are a reciprocal measure for the affinity of the enzyme to the substrate. The Km value is the substrate concentration at which the velocity (v0) of the reaction is ½ of the maximal velocity (vmax), and a low KM indicates a high substrate affinity, whereas a high KM indicates a low substrate affinity [34].

Because glucose and ketone bodies have similar Km values for glucose transport to the brain, KB is utilized as an energy source by the central nervous system when it reaches a concentration of approximately 4 mmol/L [33], which is close to the Km for the monocarboxylate transporter [35]. The energy produced by KB is higher than that produced by glucose [34,35,36,37]. The appearance of increased ketone bodies proves that they become a significant indirect source of energy and that limiting carbohydrates in the diet changes cellular metabolism. Therefore, it was expected that these changes would result in a reduction in body and fat mass.

### 4.2. KD Significantly Reduces Body Mass and Body Fat

As expected, a two-week ketogenic diet with a caloric restriction of 500 kcal/day lower than the daily energy requirement resulted in significant changes in the participants’ body composition—that is, a decrease in body mass, fat mass, and visceral fat. The results described above are similar to those obtained after a two-week restrictive ketogenic diet based on animal fat [7]. The average body mass loss was almost 3 kg vs. 2 kg, and the body fat percentage was lower by approximately 1% vs. 2% in the current research and the study conducted by Nazarewicz et al. [7]. Similar results were observed in other studies, in which low-carbohydrate diets were used for an equal period [38]. A positive effect of this diet is a reduction in visceral fat, which is associated with health benefits [39]. The diet also increased the omega-3 fatty acid concentration, which augmented the reduction in abdominal fat mass and percentage in overweight or obese individuals subjected to twelve weeks reducing diet [40]. Fish intake closely reflects *n*-3 PUFA consumption, with intakes of approximately 3–4 g/day by Inuit people, 5–6 g/day by the Japanese, and 0.25 g/day by Europeans and North Americans [41,42]. In our study, the amount of omega-3 fatty acids supplied in the diet was much higher than that in the previously cited studies, amounting to approximately 50–90 g/day, depending on the total energy demand of the subjects. The results showed that, in our study, body mass was approximately 1 kg lower after two weeks than after an analogous diet low in omega-3 fatty acids [7]. Couet et al. documented that dietary fish oil reduced body fat mass and stimulated lipid oxidation in healthy adults [43].

Moreover, Thorsdottir et al. [44] reported that after four weeks, men receiving cod, salmon, or fish oil capsules lost approximately 1 kg more than those receiving 30% caloric restriction alone. Fish species and fish oil capsules supplied with various amounts of *n*-3 PUFA did not influence body mass loss outcomes [43]. However, a limitation of this study is that we could not determine whether the obtained effects resulted from increasing the amount of omega-3 fatty acids, the composition of the diet, or caloric restriction caused by the applied diet. Considering the similarity between the effects of these studies and those of earlier cited studies [7,43,44], it cannot be excluded that the main factor determining their effectiveness is the simultaneous combination of caloric restriction and omega-3-rich diets.

Additionally, there is great interest in whether increasing the amount of omega-3 fatty acids in the diet can inhibit the loss of muscle mass following a restrictive diet. Maintaining or minimizing muscle mass loss, an adverse effect of diet, is one of the most critical challenges for scientists and athletes, who reduce their body mass before a competition.

### 4.3. KD Significantly Reduces Skeletal Muscle Mass but Does Not Change the Relative Isometric Strength of Skeletal Muscles

Current ideas say that taking omega-3 fatty acids can help to reduce substances that cause inflammation, which can lead to muscle breakdown and insulin resistance. Additionally, omega-3 might become part of muscle cell membranes and change how protein is balanced in muscles. Activating the mTOR pathways in muscles helps to make proteins, improves how the body responds to insulin, boosts the energy-producing parts of cells, and reduces NF-κB signaling. The authors also found that supplementation with omega-3 acids may positively affect lean body mass, skeletal muscle mass, and some measures of strength [45].

However, a two-week fish-based restrictive KD caused a significant decrease in muscle mass by approximately 0.7 kg (approximately 2.5%). This proves that, although the nutritional strategy presented here brings many benefits to body composition, it does not fully protect skeletal muscles under the conditions of dietary energy deficiency and must be taken into account when applying this type of dietary intervention. We have discussed the implications of muscle mass loss observed in our study. While the ketogenic diet has demonstrated efficacy in reducing fat mass without significantly impairing strength, the concurrent decline in muscle mass raises critical concerns that must be addressed.

Muscle mass is a key component of overall health, contributing to basal metabolic rate, physical functionality, and metabolic health. Even modest losses can have significant consequences, particularly in populations such as athletes, older adults, or those with chronic diseases, for whom preserving lean body mass is essential. The maintenance of strength observed in this study does not fully counteract the physiological disadvantages of reduced muscle tissue.

Loss of muscle mass during dietary interventions can lead to metabolic inflexibility, decreased exercise capacity, and reduced ability to maintain long-term weight management. Furthermore, it may increase the risk of sarcopenia in vulnerable populations. These outcomes underscore the need for dietary strategies that balance fat loss with muscle preservation.

As emphasized by Chung [46], KD could negatively affect muscle mass as a result of (1) a reduction in glycogen storage, (2) inadequate intake of essential proteins crucial for muscle composition, (3) changes in muscle energy availability due to the shift towards fat metabolism, and (4) impaired protein synthesis attributed to decreased insulin levels. However, the duration of the intervention used in the study (two-week KD) should be considered, as the phenomena described above, especially the drastic reduction in muscle glycogen, were particularly noticeable. Moreover, Cipryan et al. [47] pointed out that during the initial four weeks of the twelve-week very-low-carbohydrate high-fat KD intervention, there was a noticeable decline in participants’ muscle mass, which subsequently stabilized. The decrease in lean body mass in the subjects was similar after four and twelve weeks of KD and amounted to an average of 2.5 and 2.1 kg, respectively, possibly due to the initial reduction in total energy intake and changes in the carbohydrate proportion during the early stages of the KD intervention. As previously mentioned, calorie restriction could be critical for skeletal muscle mass. Greene et al. [48] showed that when weight-class athletes consumed an ad libitum low-carbohydrate KD, body mass (−3.26 kg) and lean mass (−2.26 kg) decreased, but the results in terms of lifting performance were comparable to those when an ad libitum normal diet was used. Other studies also confirmed that an eight-week KD during energy surplus and resistance training protocol decreased fat mass and visceral adipose tissue without decreasing lean body mass [49]. Another explanation for the effect of the fish-based restrictive KD on muscle mass may be its composition. Johnston et al. also used a calorie-restricted (−500 kcal) but a high-protein diet (~125 g protein/day consumed evenly throughout the day). They showed that after six weeks of the diet, the mean body mass loss was −3.2 ± 3.0 kg, but the change in fat-free mass was +1.5 ± 3.8 kg [50]. Kruszewski et al. [51] showed that after fifteen weeks of resistance training in the LCHF (low carbohydrates high fat) group, there was a significant decrease, and in the HCLF (high carbohydrates low fat) group, there was a significant increase in the maximal strength (measured in kg). This shows that it is worth investigating a high-protein restrictive KD rich in fish in subsequent studies.

The decrease in skeletal muscle mass was also accompanied by a decrease in the absolute average isometric strength of the extensor thigh by approximately 3%. However, in our study, isometric strength of the muscles was related to body mass and skeletal muscle mass after the diet, and no significant differences were found in the isometric strength of the skeletal muscles. For comparison, a thirty-day KD (54.8% fat, 40.7% protein, and 4.5% carbohydrates) administered to elite artistic gymnasts caused significant body mass loss and body fat without adverse effects on strength performance [52]. This allowed us to suggest the possible use of this diet in athletes to reduce their body mass during the pre-competition period. Four weeks of a non-energy-restricted ketogenic low-carbohydrate, high-fat diet also did not affect grip strength or time to fatigue, as measured with the handgrip test [53]. Other studies have also shown no significant effect of KD on muscle strength [54,55]; however, it should be noted that the subjects were trained.

Isometric tests are relatively simple to administer, pose minimal risk of injury, have high test–retest reliability, and are able to detect subtle changes in strength that dynamic tests may not be sensitive enough to detect [56].

### 4.4. The KD Is a Well-Tolerated Dietary Intervention

The KD presented here was well-tolerated by the subjects. Only two out of fourteen respondents reported increased hunger during the intervention, while four of them experienced a decrease in the feeling of hunger after the diet. This remained at the same level as the other participants. Interestingly, other feelings associated with diet were more favorable or did not change after the KD. As many as eight out of fourteen respondents declared that they had more energy after the diet, and only two showed a deterioration in this element. These studies show that the subjects tolerated the proposed two-week fish-based restrictive KD well, so it seems easy to apply. These interesting results are in contrast to those observed in individuals after a 6% body mass loss. Participants reported a significant decrease in positive mood states of vigor and an increase in negative mood states of tension, depression, anger, fatigue, and confusion [57]. Additionally, results from a randomized crossover controlled trial showed that three weeks of long-term nutritional ketosis had no effect on cognitive function, mood, or subjective sleep quality in a sample of healthy individuals [58].

To sum up, the absence of significant differences in somatic disorders following a two-week fish-based restrictive ketogenic diet (KD) suggests good tolerability. While minor adverse effects, such as bad breath and increased thirst, were reported, beneficial effects were observed in a greater number of subjects. These findings contribute to the broader discussion on ketogenic diet tolerability. It is also worth mentioning the study conducted by Arab et al. [59] focusing on a systematic review assessing various dietary patterns, including the ketogenic diet, in relation to mood regulation. Their findings suggest that the ketogenic diet may have mood-stabilizing effects by improving vigor, reducing fatigue, and stabilizing blood glucose levels. However, this study also acknowledges inconsistencies across included research, limiting definitive conclusions on long-term adherence and tolerability. Dietch et al. [60] also described the relationship between ketogenic and low-carbohydrate diets and mood and anxiety disorders, noting that transient side effects such as gastrointestinal discomfort, fatigue, and keto-adaptation symptoms can vary among individuals. This aligns with the reported adverse effects in the current study, reinforcing the importance of further research comparing different ketogenic diet formulations.

### 4.5. KD—Future Research Perspectives

To better understand the potential advantages of the KD, future studies could directly compare the tolerability of a no-meat ketogenic diet with that of a standard ketogenic diet. Such comparisons could help to determine if the elimination of meat (other than fish) contributes to improved tolerance and fewer adverse effects, which would be beneficial for populations seeking to adopt a ketogenic diet with minimal discomfort. Additionally, a broader sample size and longer duration of study could provide more robust data on this topic.

A direct comparison between the fish-based restrictive KD and standard ketogenic diets would be valuable in determining whether this specific variation offers improved adherence and metabolic outcomes. Given that tolerability can be influenced by macronutrient composition and individual metabolic responses, future research should explore long-term effects, subjective well-being, and physiological adaptations to optimize clinical recommendations. In comparing dietary approaches, a ketogenic diet with a caloric deficit of 500 kcal may be advantageous for preserving lean mass due to its higher protein content and fat utilization. Research indicates that ketogenic diets can reduce muscle loss during caloric restriction more effectively than standard diets by promoting fat oxidation and maintaining metabolic efficiency [61], but on the other hand, there is some evidence that in untrained individuals, the ketogenic diet may lead to slightly higher lean body mass loss compared to a normal diet [62].

Future research should focus on integrating different types of training or adjusting macronutrient ratios, such as increasing protein intake, to counteract muscle loss during ketogenic interventions. By addressing these concerns, dietary protocols can be optimized to maximize fat loss while preserving or enhancing muscle mass, ensuring better health and functional outcomes for diverse populations.

### 4.6. Study Limitations

This study has some potential limitations. The present study includes a relatively short observation period. Although the duration of the diet, i.e., 14 days, may raise concerns, it should be noted that the choice of this period was deliberate, as it corresponds to the period of body mass reduction in an athlete’s preparation for a competition. However, it would be worth conducting research taking into account a longer period, after which the effects obtained in this work could turn out to be even stronger. Furthermore, no control group undergoing no caloric restriction or a KD based on other products excluding fish was analyzed, which would also be worth considering in future studies.

This study’s small sample size should also be acknowledged. However, our previously published findings on a 2-week KD [7] suggest that a sample size of 12 is sufficient, indicating that the observed effect sizes were significant. Nevertheless, further studies with more subjects and in a population with a different diet than in Poland, where fish consumption is not high, would also be extremely valuable.

The strength of this study is that it was conducted on a restrictive ketogenic diet that does not contain meat, which is one of the most important and most frequently consumed foodstuffs in the diet for most people on Earth. Moreover, these studies have shown that obtaining beneficial effects in the form of weight loss and body fat is possible with a diet rich in fish without significant adverse changes in the subjects’ mood.

## 5. Conclusions

The implementation of a fish-based restrictive KD resulted in favorable changes in body composition, including reductions in total body mass and visceral fat. Importantly, although skeletal muscle mass decreased by approximately 2%, relative isometric muscle strength remained unaffected, supporting the hypothesis that muscle functionality can be preserved under such dietary conditions. Furthermore, participants demonstrated good tolerance to the ketogenic diet, with no significant somatic disorders observed during the study period. These findings suggest that this dietary approach may serve as an effective intervention for achieving rapid reductions in body mass among healthy adults, particularly athletes, without impairing muscle strength. This aligns with the study’s objectives to evaluate the effects of a modified ketogenic diet on body composition, strength capacity, and somatic health.

## Figures and Tables

**Figure 1 nutrients-17-01297-f001:**
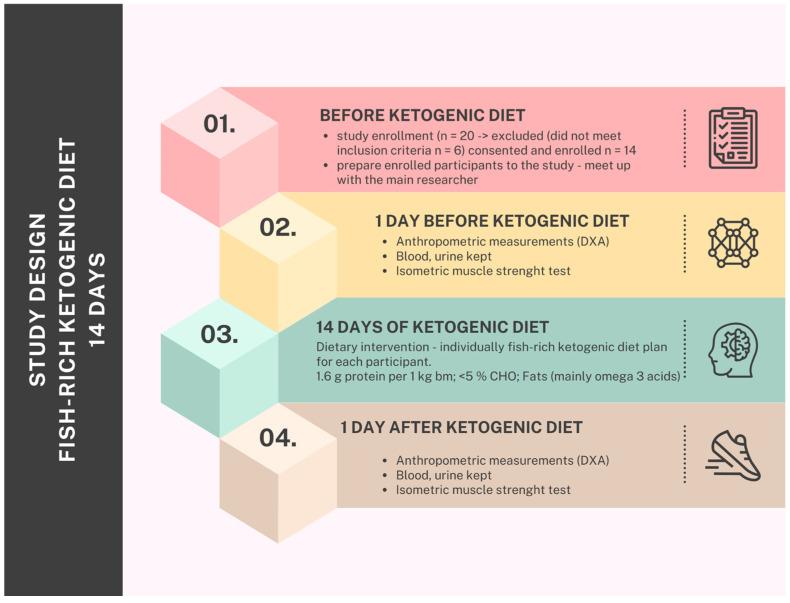
Study design.

**Figure 2 nutrients-17-01297-f002:**
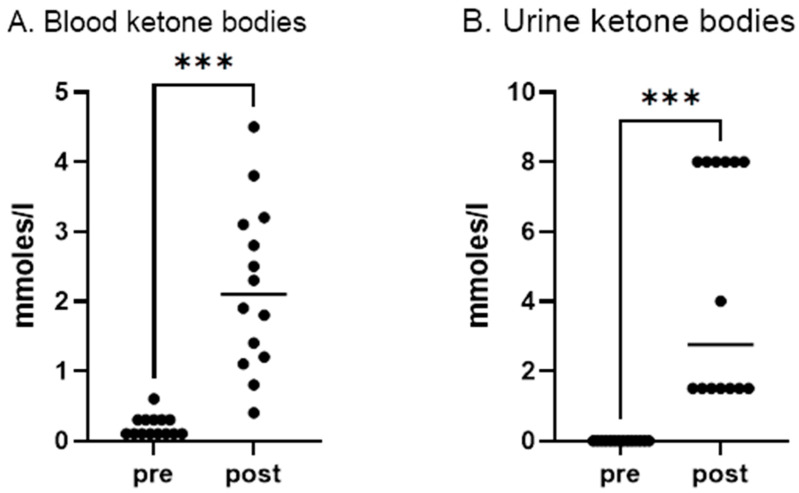
The concentration of blood and urine ketone bodies in subjects before (PRE) and after (POST) a 2-week fish-based restrictive ketogenic diet. Concentration of (**A**) ketone bodies in the blood (***—*p* < 0.001) and (**B**) urine (***—*p* < 0.001) in subjects before (PRE) and after (POST) a 2-week meatless restrictive ketogenic diet. Values are expressed as mean ± SD. *p* < 0.05 was considered statistically significant.

**Figure 3 nutrients-17-01297-f003:**
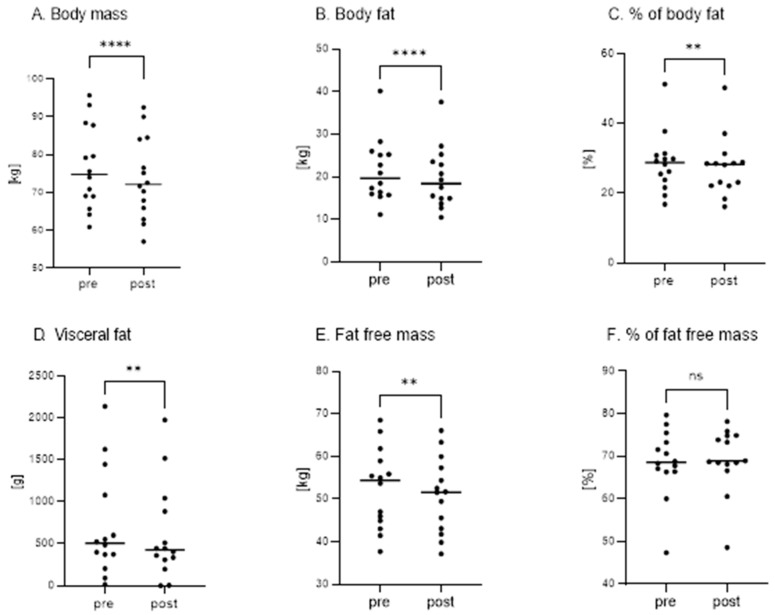
Body composition parameters in adult participants before (PRE) and after (POST) a 2-week fish-based restrictive ketogenic diet. (**A**) Body mass (BM) (****—*p* < 0.001), (**B**) body fat (BF) (****—*p* < 0.001), (**C**) percentage of body fat (BF%,) (**—*p* = 0.0019), (**D**) visceral fat (VF) (**—*p* = 0.0012), (**E**) fat-free mass (FFM) (**—*p* = 0.0017), and (**F**) percentage of fat-free mass (FFM%) (ns—not significant) in adult participants before (PRE) and after (POST) a 2-week fish-based restrictive ketogenic diet. Values are expressed as mean ± SD. *p* < 0.05 was considered statistically significant.

**Figure 4 nutrients-17-01297-f004:**
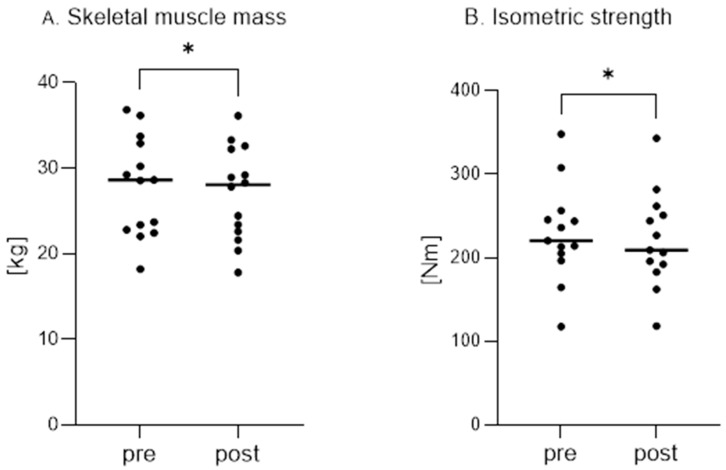
Skeletal muscle mass and the average value of isometric strength of the extensor muscles of the thigh in subjects before (PRE) and after (POST) a 2-week fish-based restrictive ketogenic diet. (**A**) Skeletal muscle mass (*—*p* = 0.01) and (**B**) the average value of isometric strength of the extensor muscles of the thigh (*—*p* = 0.0256) in subjects before (PRE) and after (POST) a 2-week fish-based restrictive ketogenic diet. Values are expressed as mean ± SD. *p* < 0.05 was considered statistically significant.

**Table 1 nutrients-17-01297-t001:** Flexor and extensor peak and average knee torque expressed per 1 kg of body mass (BM) before (PRE) and after (POST) 2-week KD in healthy adults.

	PRE	POST	*p*
Flexor peak (Nm/kg BM)	1.27 ± 0.34	1.28 ± 0.35	0.652
Flexor average (Nm/kg BM)	1.19 ± 0.30	1.20 ± 0.33	0.603
Extensor peak (Nm/kg BM)	3.08 ± 0.70	3.01 ± 0.57	0.753
Extensor average (Nm/kg BM)	2.98 ± 0.67	3.00 ± 0.64	0.553

The flexor and extensor peak and average knee torque were expressed per 1 kg of body mass (BM) before (PRE) and after (POST) the 2-week KD in healthy adults (*n* = 14). Values are expressed as mean ± SD. *p* < 0.05 was considered statistically significant.

**Table 2 nutrients-17-01297-t002:** Flexor and extensor peak and average knee torque expressed per 1 kg of skeletal muscle mass (SMM) before (PRE) and after (POST) 2-week KD in healthy adults.

	PRE	POST	*p*
Flexor peak (Nm/kg SMM)	3.46 ± 0.73	3.43 ± 0.72	0.647
Flexor average (Nm/kg SMM)	3.25 ± 0.66	3.23 ± 0.68	0.756
Extensor peak (Nm/kg SMM)	8.42 ± 1.30	8.13 ± 1.08	0.274
Extensor average (Nm/kg SMM)	8.14 ± 1.29	8.08 ± 1.16	0.661

Flexor and extensor peak and average knee torque expressed per 1 kg of skeletal muscle mass before (PRE) and after (POST) the 2-week KD in healthy adults (*n* = 14). Values are expressed as mean ± SD. *p* < 0.05 was considered statistically significant.

**Table 3 nutrients-17-01297-t003:** Analysis of participants’ somatic disorders before (PRE) and after (POST) 2-week KD.

Somatic Disorders	PRE	POST	Improvement	Deterioration
Drowsiness	2.36 ± 1.50	1.57 ± 1.16	5/14	0/14
Energy	3.14 ± 1.29	3.50 ± 1.56	8/14	2/14
Nausea	1.64 ± 1.15	1.64 ± 1.34	3/14	2/14
Vomiting	1.00 ± 0.00	1.00 ± 0.00	0/14	0/14
Concentration disorders	2.57 ± 1.55	1.79 ± 1.25	5/14	0/14
Sleeping disorders	1.29 ± 0.61	1.50 ± 1.16	2/14	2/14
Diarrhea	1.00 ± 0.00	1.00 ± 0.00	0/14	0/14
Constipation	3.07 ± 1.64	2.64 ± 1.86	4/14	1/14
Bad breath	2.29 ± 1.38	2.43 ± 1.60	1/14	2/14
Increased thirst	3.57 ± 1.65	4.14 ± 1.17	1/14	4/14
Hunger	2.50 ± 1.61	2.21 ± 1.53	4/14	2/14

## Data Availability

All data generated or analyzed during this study are included in this article. Further enquiries can be directed to the corresponding author.

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
