# Peer review of "Impact of a Fish-Based Restrictive Ketogenic Diet on Body Composition and Strength Capacity: A Pre–Post Study"

_nutrients, 2025, doi:10.3390/nu17081297_

Round 1
Reviewer 1 Report
Comments and Suggestions for Authors
Dear Authors,
Thank you for submitting this manuscript regarding KD. I believe the topic addressed aligns well with the journal's objectives. Below are my comments and suggestions:
- TITLE: Please include the study design in the title.
- ABSTRACT: In this section, I would also request that the study design be described more clearly.
- INTRODUCTION:
The introduction is clear, but I would suggest that the authors improve the flow between paragraphs by adding connecting phrases. - Additionally, in lines 73-76, when discussing the important role of omega-3 in reducing inflammation, a clinical link could be made, as done in the following paragraph, to specific conditions, such as the reduction of inflammation in "oncological treatments," but also, for instance, to recent studies on the "effect of omega-3 in patients undergoing bone marrow transplantation"
- METHODS:
The methodology appears solid. However, I recommend specifying that the study followed a specific reporting checklist (you may refer to the Equator Network guidelines). - RESULTS and DISCUSSION:
In the results section, I suggest describing the sample. While this is done in the methods section, here it should focus only on the methodology for enrollment, while the sample itself should be described.
The text in both the results and discussion sections is well-written and clear. However, the discussion section appears quite lengthy. I appreciated the subdivision into subheadings but suggest that the authors reduce the length of the discussion. Perhaps consider removing future perspectives from the various sections and, at the end of the discussion, create a subsection on study limitations and another one on future research perspectives. - REFERENCES:
Please align the references according to the journal’s guidelines.
Author Response
Comments 1: TITLE: Please include the study design in the title.
Response 1: Thank you for this comment. We've included the suggestions you submitted. Impact of a Fish-Based Restrictive Ketogenic Diet on Body Composition and Strength Capacity: A Pre-Post Study
Comments 2: ABSTRACT: In this section, I would also request that the study design be described more clearly.
Response 2: Thank you for this comment. We've included the suggestions you submitted. The study was designed and conducted as a single-group pre-post study [lines 22-23].
Comments 3: INTRODUCTION: The introduction is clear, but I would suggest that the authors improve the flow between paragraphs by adding connecting phrases.
Response 3: Thank you for this comment. We've included the suggestions you submitted.
Comments 4: Additionally, in lines 73-76, when discussing the important role of omega-3 in reducing inflammation, a clinical link could be made, as done in the following paragraph, to specific conditions, such as the reduction of inflammation in "oncological treatments," but also, for instance, to recent studies on the "effect of omega-3 in patients undergoing bone marrow transplantation"
Response 4: Thank you for this comment. We've included the suggestions you submitted [lines 99-102]. Moreover, omega-3 has been clinically linked to reducing inflammation in oncological treatments [18] and has shown promise in recent studies on the effect of omega-3 in patients undergoing bone marrow transplantation [19].
Comments 5: METHODS: The methodology appears solid. However, I recommend specifying that the study followed a specific reporting checklist (you may refer to the Equator Network guidelines).
Response 5: Thank you for this comment. We've included the suggestions you submitted [lines 154-175].
2.3. Eligibility Criteria
The inclusion criteria were female or male, non-smoker, drinking alcohol occasionally, and not subjected to physical exercise for at least 48 h before examination. Exclusion criteria included cardiovascular, thyroid, gastrointestinal, respiratory, or other metabolic diseases; adherence to special diets; nutritional supplements; and medication to control blood lipids or glucose. During the study protocol, the participants were asked to stop physical exercises. While the study was conducted, no specific cultural or geographical dietary influences were identified that could impact the findings.
2.4. Ethics
All subjects read and signed an informed consent document with the description of the testing procedures approved by the Independent Bioethics Committee for Scientific Research, District Medical Chamber in Gdansk (KB 27-19), and conformed to the standards for the use of human subjects in research, as outlined in the Declaration of Helsinki, Clinical Trial registration number NCT05558488.
2.5. Study Design and Procedures
This study was designed as a one-sample pre-post study, where the same group of participants was assessed before and after a 14-day dietary intervention. This design allows for evaluating within-subject changes over time, minimizing inter-individual variability. By comparing baseline and post-intervention measurements, the study aims to determine the effects of a fish-based ketogenic diet. The pre-post approach is particularly useful for short-term dietary interventions, as it provides insights into rapid physiological adaptations without the need for a control group.
Comments 6: RESULTS and DISCUSSION: In the results section, I suggest describing the sample. While this is done in the methods section, here it should focus only on the methodology for enrollment, while the sample itself should be described.
Response 6: Thank you for this comment [lines 143-153]
2.2. Participants
The study was conducted with recruitment carried out through advertisements, posters displayed at the university, and information shared among colleagues. Efforts were made to ensure diversity among participants, resulting in a group with varying body weight, genders, and levels of physical activity. Fourteen healthy participants (age 45 ± 8 years; height 174 ± 10 cm; body mass 76.7 ± 11.1 kg, female n = 5, male n = 9) were recruited for the study by self-selection (20 people were recruited, and finally 14 people were enrolled in the study). Data was collected for all participants (n = 14), except isometric muscle strength (n = 13). The participants included both active and sedentary individuals, and the effects of the ketogenic diet were observed consistently across both groups, enhancing the applicability of the results to a wider population.
Comments 7: The text in both the results and discussion sections is well-written and clear. However, the discussion section appears quite lengthy. I appreciated the subdivision into subheadings but suggest that the authors reduce the length of the discussion. Perhaps consider removing future perspectives from the various sections and, at the end of the discussion, create a subsection on study limitations and another one on future research perspectives.
Response 7: Thank you for this comment. We've included the suggestions you submitted [lines 558-602]
4.5. KD - future research perspectives
To better understand the potential advantages of KD, future studies could directly compare the tolerability of a no-meat ketogenic diet with that of a standard ketogenic diet. Such comparisons could help determine if the elimination of meat (other than fish) contributes to improved tolerance and fewer adverse effects, which would be beneficial for populations seeking to adopt a ketogenic diet with minimal discomfort. Additionally, a broader sample size and longer duration of study could provide more robust data on this topic.
A direct comparison between the fish-based restrictive KD and standard ketogenic diets would be valuable in determining whether this specific variation offers improved adherence and metabolic outcomes. Given that tolerability can be influenced by macronutrient composition and individual metabolic responses, future research should explore long-term effects, subjective well-being, and physiological adaptations to optimize clinical recommendations. In comparing dietary approaches, a ketogenic diet with a caloric deficit of 500 kcal may be advantageous for preserving lean mass due to its higher protein content and fat utilization. Research indicates that ketogenic diets can reduce muscle loss during caloric restriction more effectively than standard diets by promoting fat oxidation and maintaining metabolic efficiency [61], but on the other hand, there is some evidence that in untrained individuals, the ketogenic diet may lead to slightly higher lean body mass loss compared to a normal diet [62].
Future research should focus on integrating different types of training or adjusting macronutrient ratios, such as increasing protein intake, to counteract muscle loss during ketogenic interventions. By addressing these concerns, dietary protocols can be optimized to maximize fat loss while preserving or enhancing muscle mass, ensuring better health and functional outcomes for diverse populations.
4.6. Study limitations
The study has potential limitations. The present study includes a relatively short observation period. Although the duration of the diet, i.e. 14 days, may raise concerns, it should be noted that the choice of this period was deliberate, as it corresponds to the period of body mass reduction in preparation of the athlete for the competition. However, it would be worth conducting research taking into account a longer period, after which the effects obtained in this work could turn out to be even stronger. Furthermore, no control group undergoing no caloric restriction or KD based on products, excluding fish, was analyzed, which would also be worth considering in future studies.
The study's small sample size should also be acknowledged. However, our previously published findings on a 2-week KD [7] suggest that a sample size of 12 was sufficient, indicating that the observed effect sizes were significant. Nevertheless, further studies with more subjects and in a population with a different diet than in Poland, where fish consumption is not high, would also be extremely valuable.
The strength of this study is that it was conducted on a restrictive ketogenic diet that does not contain meat, which is one of the most important and most frequently consumed nutrients in the diet for most people on Earth. Moreover, these studies have shown that obtaining beneficial effects in the form of weight loss and body fat is possible with a diet rich in fish without significant adverse changes in the subjects' mood.
Comments 8: REFERENCES: Please align the references according to the journal’s guidelines.
Response 8: Thank you for this comment. We've included the suggestions you submitted. The bibliography has been revised.
Thank you very much for all your valuable comments and suggestions, which will significantly help improve this manuscript.
Reviewer 2 Report
Comments and Suggestions for Authors
Interesting manuscript entitled: Impact of a Fish-Based Restrictive Ketogenic Diet on Body 2 Composition and Strength Capacity
I have some suggestions on how to improve the document:
1. between line 72 and 80 describe the properties of EPA and DHA, as they are generally different properties, I suggest describing one first, then the properties of the other fatty acid.
2. At the end of the introduction, add the objective of the study.
3. Describe the type of study
4. in the figures, in particular in figure 3, add what is being assessed, as the letters that identify (e.g. A. body mass).
5. in the discussion, on line 351, reference the idea.
6. when the dietary recommendations were made, were there any types of fish that were recommended or was it just fish?
7. which professional carried out the dietary analysis, was he/she trained in the area, please mention
8. check the format of the references, it is not correct.
Comments on the Quality of English Language
Interesting manuscript entitled: Impact of a Fish-Based Restrictive Ketogenic Diet on Body 2 Composition and Strength Capacity
I have some suggestions on how to improve the document:
1. between line 72 and 80 describe the properties of EPA and DHA, as they are generally different properties, I suggest describing one first, then the properties of the other fatty acid.
2. At the end of the introduction, add the objective of the study.
3. Describe the type of study
4. in the figures, in particular in figure 3, add what is being assessed, as the letters that identify (e.g. A. body mass).
5. in the discussion, on line 351, reference the idea.
6. when the dietary recommendations were made, were there any types of fish that were recommended or was it just fish?
7. which professional carried out the dietary analysis, was he/she trained in the area, please mention
8. check the format of the references, it is not correct.
Author Response
Comments 1: between line 72 and 80 describe the properties of EPA and DHA, as they are generally different properties, I suggest describing one first, then the properties of the other fatty acid.
Response 1: Thank you for this comment. We've included the suggestions you submitted [lines 80-93]
EPA is a polyunsaturated fatty acid with a 20-carbon chain and five cis double bonds. EPA is known for its anti-inflammatory properties, acting as a precursor for prostaglandin-3, thromboxane-3, and leukotriene-5 eicosanoids, which play roles in reducing inflammation and preventing blood clotting. It is primarily found in cold-water fish like salmon, mackerel, and sardines, and can also be obtained from fish oil supplements. EPA is used to lower triglyceride levels in the blood, reduce the risk of heart disease, and alleviate symptoms of depression [15].
DHA is another polyunsaturated fatty acid, but with a 22-carbon chain and six cis double bonds. DHA is crucial for brain health, supporting cognitive function and eye health. It is a major component of neuronal membranes and helps in the optimal functioning of neuronal membrane proteins. DHA is found in fatty fish like salmon, herring, and mackerel, as well as in fish oil and algae oil. It is important for brain development, particularly in infants, and also supports heart health, potentially reducing the risk of chronic diseases [16]
Comments 2: At the end of the introduction, add the objective of the study.
Response 2: Thank you for this comment. We've included the suggestions you submitted [lines 128-134].
This study aims to fill this gap by examining the effects of a fish-based restrictive KD on body mass, body composition, and skeletal muscle strength over a 14-day period, providing insight into its potential benefits for athletes and individuals seeking rapid body mass reduction without significant muscle loss. This study aimed to assess the effects of a two-week fish-based restrictive KD on body composition, strength capacity (isometric muscle strength), and somatic disorders in healthy adults.
Comments 3: Describe the type of study
Response 3: Thank you for this comment. We've included the suggestions you submitted [lines 22-23].
Comments 4: in the figures, in particular in figure 3, add what is being assessed, as the letters that identify (e.g. A. body mass).
Response 4: Thank you for this comment. We've included the suggestions you submitted.
Comments 5: in the discussion, on line 351, reference the idea.
Response 5: Thank you for this comment. There is no citation here – these are the results from our experiment.
Comments 6: when the dietary recommendations were made, were there any types of fish that were recommended or was it just fish?
Response 6: Thank you for this comment. The study participants had a prepared dietary plan (caloric intake individually adjusted according to the methodology description). The fish included in the plan were cod, mackerel, pollock, tuna, and salmon.
Comments 7: which professional carried out the dietary analysis, was he/she trained in the area, please mention
Response 7: Thank you for this comment. The person who carried out the dietary analysis was someone who has completed both bachelor's and master's degrees in the field of dietetics.
Comments 8: check the format of the references, it is not correct.
Response 8: Thank you for this comment. We've included the suggestions you submitted. The bibliography has been revised.
Thank you very much for all your valuable comments and suggestions, which will significantly help improve this manuscript.
Round 2
Reviewer 1 Report
Comments and Suggestions for Authors
the authors have provided appropriate modifications.